

# Measuring language lateralisation with different language tasks: a systematic review

Abigail R. Bradshaw, Paul A. Thompson, Alexander C. Wilson, Dorothy V.M. Bishop and Zoe V.J. Woodhead

Department of Experimental Psychology, University of Oxford, Oxford, United Kingdom

## ABSTRACT

Language lateralisation refers to the phenomenon in which one hemisphere (typically the left) shows greater involvement in language functions than the other. Measurement of laterality is of interest both to researchers investigating the neural organisation of the language system and to clinicians needing to establish an individual's hemispheric dominance for language prior to surgery, as in patients with intractable epilepsy. Recently, there has been increasing awareness of the possibility that different language processes may develop hemispheric lateralisation independently, and to varying degrees. However, it is not always clear whether differences in laterality across language tasks with fMRI are reflective of meaningful variation in hemispheric lateralisation, or simply of trivial methodological differences between paradigms. This systematic review aims to assess different language tasks in terms of the strength, reliability and robustness of the laterality measurements they yield with fMRI, to look at variability that is both dependent and independent of aspects of study design, such as the baseline task, region of interest, and modality of the stimuli. Recommendations are made that can be used to guide task design; however, this review predominantly highlights that the current high level of methodological variability in language paradigms prevents conclusions as to how different language functions may lateralise independently. We conclude with suggestions for future research using tasks that engage distinct aspects of language functioning, whilst being closely matched on non-linguistic aspects of task design (e.g., stimuli, task timings etc); such research could produce more reliable and conclusive insights into language lateralisation. This systematic review was registered as a protocol on Open Science Framework: https://osf.io/5vmpt/.

Corresponding author
Abigail R. Bradshaw,
abigail.bradshaw@psy.ox.ac.uk

## INTRODUCTION

It is well established that for most individuals, the left hemisphere is dominant in mediating language functions, as proposed in the 19th century by Paul Broca. Our understanding of such hemispheric specialisation for language in the centuries since still leaves many unanswered questions. Because both expressive and receptive aphasia are so reliably associated with left-hemisphere injury, there tends to be an assumption that left-sided lateralisation is a general feature of language processing, consistent across language

**Table 1  Model-based predictions of language lateralisation.** The table illustrates some predictions of different models of the neural basis of language, in terms of the lateralisation expected for different aspects of language processing.

| Theoretical principle/ Model | Speech acoustic processing | Speech comprehension | Speech articulation | Semantics | Syntax |
|---|---|---|---|---|---|
| Dual stream model of speech processing (*Hickok & Poeppel, 2007*) | B | B | L | B | – |
| Hierarchical asymmetry of linguistic complexity (*Peelle, 2012*) | B | L | – | L | L |
| Bilateral sensorimotor inputs/outputs and left lateralised central language processes (*Price, 2012*) | B | L | B | L | L |
| COM-PRE hypothesis (*Poeppel, 2014*) | B | L | B | L | L |

Notes.
B, Bilateral; L, Lateralised.

domains. Nevertheless, there is evidence that lateralisation may differ within individuals for different language functions, as well as between individuals in side and extent.

Early suggestions of within-individual variability can be found in *Rasmussen & Milner*'s (*1975*) accounts of Wada testing in patients undergoing surgery to treat epilepsy. They reported on several patients with bilateral speech representation, manifest as a dissociation between the hemispheric organisation of different language functions. Specifically, while anaesthetic injection to one hemisphere selectively disrupted naming and not verbal serial order tasks (e.g., reciting the days of the week), an injection to the other hemisphere produced the reverse pattern. This was construed as evidence that in some cases a 'division of labour' can exist between the hemispheres, in which different 'speech centres' can lateralise to different hemispheres independently. Although such evidence was from the study of a special population, it was argued that such a phenomenon should not necessarily be considered as a result of the type of brain damage and reorganisation that occurs in epilepsy. This raises the possibility that cerebral lateralisation may be a multifactorial rather than a unitary process, with different language processes developing hemispheric lateralisation independently, and to varying degrees (*Bishop, 2013*).

Indeed, several contemporary models of language predict different patterns of lateralisation for different language processes (*Hickok & Poeppel, 2007*; *Peelle, 2012*; *Poeppel, 2014*; *Price, 2012*). These predictions are summarised in Table 1. Different models make different distinctions between language processes and use different terminology, but some general patterns emerge. Acoustic processing of speech input and speech articulation are generally considered to be bilateral, whereas comprehension and generation of more meaningful language is considered to be lateralised. There are some points of disagreement between theories however, either in terms of the extent of lateralisation for a particular language process or the theoretical reasons proposed for such patterns of lateralisation.

Contemporary non-invasive techniques allow more extensive research on patterns of laterality than earlier clinical studies. Functional magnetic resonance imaging (fMRI) data can be used to calculate a laterality index (LI), a single value description of the predominance of activity in one hemisphere. The LI is calculated as the difference between

activity in each hemisphere (L and R) divided by the total activity across the hemispheres.

$$LI = \frac{L-R}{L+R}.$$

Multiple language tasks have been used with fMRI. At first glance, the literature appears to support the notion that language laterality is not unitary, because we can see differences between tasks in the strength of the laterality measurements they yield. However, the reasons for such variability in LI strength across language tasks can be debated; could it simply be an artefact of more trivial differences in task design, or does it reveal something fundamental about the hemispheric organisation of different components of language? Of course, trying to devise tasks so as to equate diverse language functions such as speech production and speech comprehension is an unrealistic and inappropriate goal. However, more can be done to optimize protocols for LI measurement, in order to try to reduce the possibility of differences in task sensitivity or measurement error being responsible for variability in LIs across tasks.

This systematic review aims to assess evidence on the robustness of laterality measured using fMRI with different language tasks, from studies published between 2000 and 2016. This is done with a view to providing some guidance on optimizing variables such as region of interest and baseline task on a task-by-task basis. Such optimization will be important before tasks can be used to systematically probe patterns of co-lateralisation and independent lateralisation of different language functions. We hypothesise that (1) different language tasks will demonstrate different levels of lateralisation and (2) parameters such as the region of interest and baseline task used will have effects on laterality measurement that may be task-specific.

## MATERIALS AND METHODS

A protocol for this systematic review has been registered on Open Science Framework and can be found at https://osf.io/5vmpt/. We do not cover here generic issues such as how thresholding and other methodological issues affect laterality measurement, since these were the focus of a companion review based on the same source material (*Bradshaw, Bishop & Woodhead, 2017*).

### Eligibility criteria

We selected papers published between 2000 and 2016 that used fMRI to study language lateralisation and that met the following criteria: (1) the paper reported LIs for language calculated using fMRI; (2) the paper studied healthy monolingual adults; and (3) if both patients and healthy controls were studied, the data for controls were reported separately. Papers were excluded if: (1) they exclusively studied structural asymmetries, children or bilingualism; or (2) they used language tasks with non-European languages. The search was restricted to studies of healthy, monolingual, adult participants to reduce heterogeneity within our study sample.

### Search strategy and selection process

The following search terms were used to search papers published between 2000 and 2016 in Web of Science: laterali* OR asymmetr* OR dominance; AND language OR reading; AND

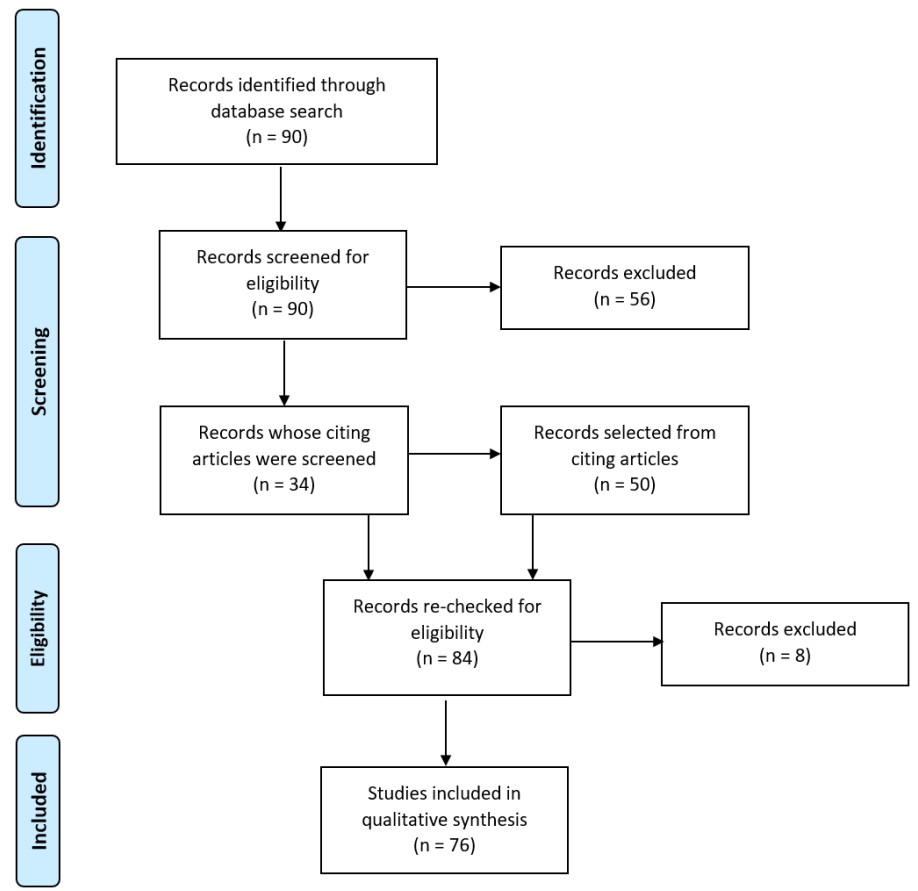

**Figure 1** **Literature search and selection process.** Flow diagram illustrating the search and selection process for obtaining papers for inclusion in the review. Adapted from *Moher et al. (2009)*.

fMRI OR functional MRI OR functional magnetic resonance imaging OR functional MR OR function MRI; NOT schizophrenia; NOT development*; NOT child*; NOT bilingual*. This was last searched on 05/12/16. Titles and abstracts of the resulting 90 papers were then screened by two of the review authors (Abigail Bradshaw and Zoe Woodhead), followed by full-text scans to determine whether the inclusion criteria were met. Selected lists were compared between reviewers and any discrepancies discussed and a mutual decision made. This resulted in the selection of 34 papers. We next screened papers citing these 34 articles. A further 50 articles were identified as meeting our criteria, yielding a total of 84 papers. A final check of papers led to the discounting of 7 papers deemed to not sufficiently meet criteria, with a further paper being discounted during conductance of the review. The full search and selection process is illustrated in Fig. 1. A list of the 76 selected papers is given in Appendix S1.

## Data collection and analysis

Information on variables of interest for each study were collected and managed using REDCap electronic data capture tools (*Harris et al., 2009*) hosted at Oxford University.

REDCap (Research Electronic Data Capture) is a secure, web-based application designed to support data capture for research studies, providing: (1) an intuitive interface for validated data entry; (2) audit trails for tracking data manipulation and export procedures; (3) automated export procedures for seamless data downloads to common statistical packages; and (4) procedures for importing data from external sources. The full database can be found in Appendix S2. A summary table drawn from this database with the key outcomes of interest for this paper is provided in Appendix S3. For each paper, we recorded: sample size and handedness, the type of fMRI design used, the activity measures used for LI calculation, the threshold level chosen, the use of global or regional LI calculation, the specific regions considered, the language and baseline tasks used, the use of a single or a combined task analysis and the task difficulty.

The variable nature of the methods and measures reported by different papers did not permit performance of a meta-analysis. Instead, to illustrate the strength of laterality measured across different language tahe sks, we produced forest plots showing the mean and 95% confidence intervals of LI values reported in the studies, as well as their associated methods of LI calculation, region(s) of interest, and language and baseline tasks (Figs. 2–4). However, these outcome measures were not always available in every paper; LI values and/or their spread were sometimes omitted altogether or given in a different form e.g., median values. Where standard deviation or standard error were given, these were converted to 95% confidence intervals. A spreadsheet of the data that was used to generate these forest plots in given in Appendix S4.

To avoid the potential confound of heterogeneity in samples in terms of handedness, these forest plots only included mean LIs reported by our selected studies measured either from right handed participants, or from mixed handedness samples where the relative proportion of left and right handers was representative of the general population (around 10% left handed, 90% right handed). We excluded LIs reported from studies that selected a participant group on the basis of their pre-known lateralisation. Where more than one frontal LI was reported from a study, inferior frontal gyrus (IFG) LIs were selected; where more than one temporoparietal LI was reported, the LI calculated from the largest area of temporoparietal cortex was selected. Forest plots were created using a script in R, which is available along with the data on open science framework (https://osf.io/7s4hv/).

## RESULTS

The main language tasks identified in our search are listed in Table 2, with counts of the number of studies using each task (one study is missing from these counts as their language task did not fit in to any of these categories). Mean LIs reported from studies using these different language tasks are given in Figs. 2–4. A single language task typically engages multiple language processes in an overlapping fashion. This may either be because of task requirements, or reflect spontaneous engagement of task irrelevant processing by the perception of linguistic stimuli. Table 2 provides one characterisation of the different language processes engaged by each of the language and baseline tasks included in this review. Comparing the language processes engaged by active and baseline tasks is crucial,

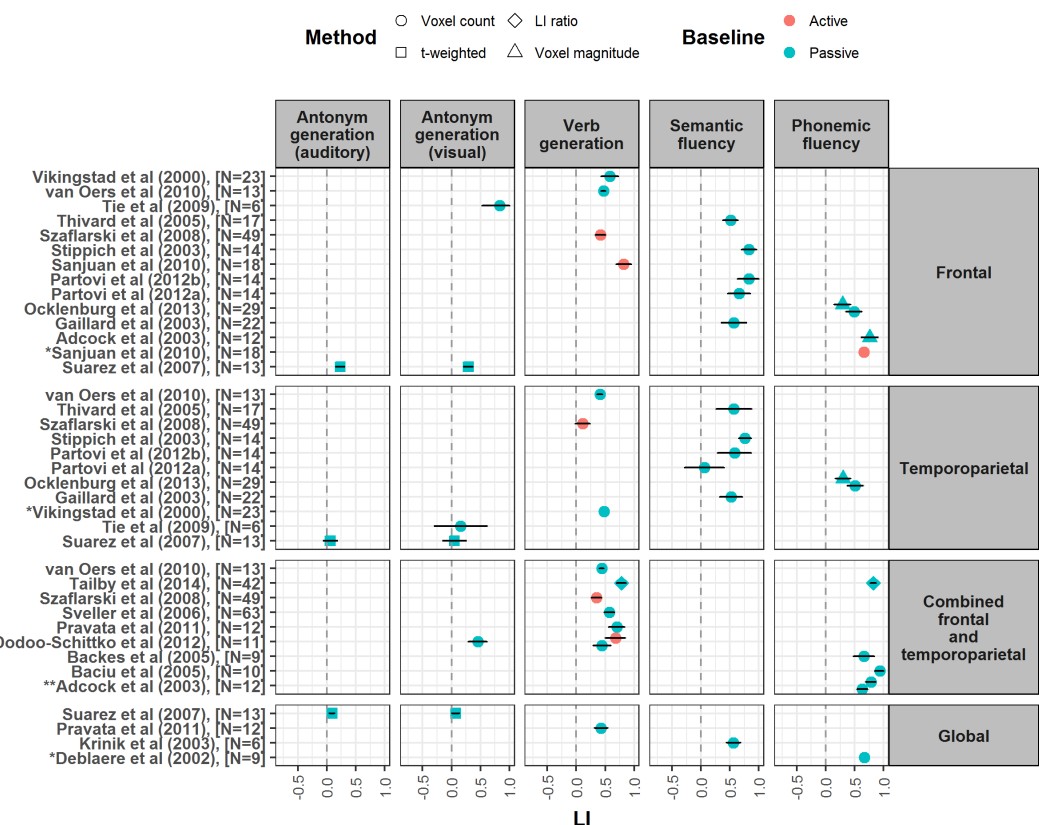

**Figure 2** **Forest plot shows mean LI values for verbal fluency tasks reported from studies meeting our criteria.** Plot is divided up according to region of interest used for LI calculation (frontal, temporoparietal, combined frontal and temporoparietal and global). Error bars represent 95% confidence limits. Colour of symbol indicates type of baseline task used (active or passive), and shape of symbol indicates method of LI calculation (see key). *Papers did not report a measure of spread for LI values, so confidence interval is not shown. ** LI values reported by this paper are given at different thresholds: $Z = 5.3$ (Top), $Z = 2.3$ (Bottom). Figures 2–4 are published on Figshare and can be found at: https://figshare.com/articles/ Forrest_Plots_of_LI_values_for_different_language_tasks/4977950.

because when activation for a language task is subtracted from a baseline the aim is to isolate specific linguistic functions, and the extent to which this is successful will depend on the demands of the baseline task.

In the following review, we discuss each language task in turn, with reference to the involvement of different language processes and the forest plots of mean LI values (Figs. 2–4). Table 2 highlights the difficulty in designing a task which isolates a single language function in order to study its laterality; this must be kept in mind when interpreting LI values and theorising on the lateralisation of particular language processes.

## Verbal fluency

Verbal fluency tasks have traditionally been viewed as the gold standard for measuring language lateralisation with fMRI. Here, the participant must generate (covertly or overtly) words that meet certain criteria, such as beginning with a particular letter (phonemic

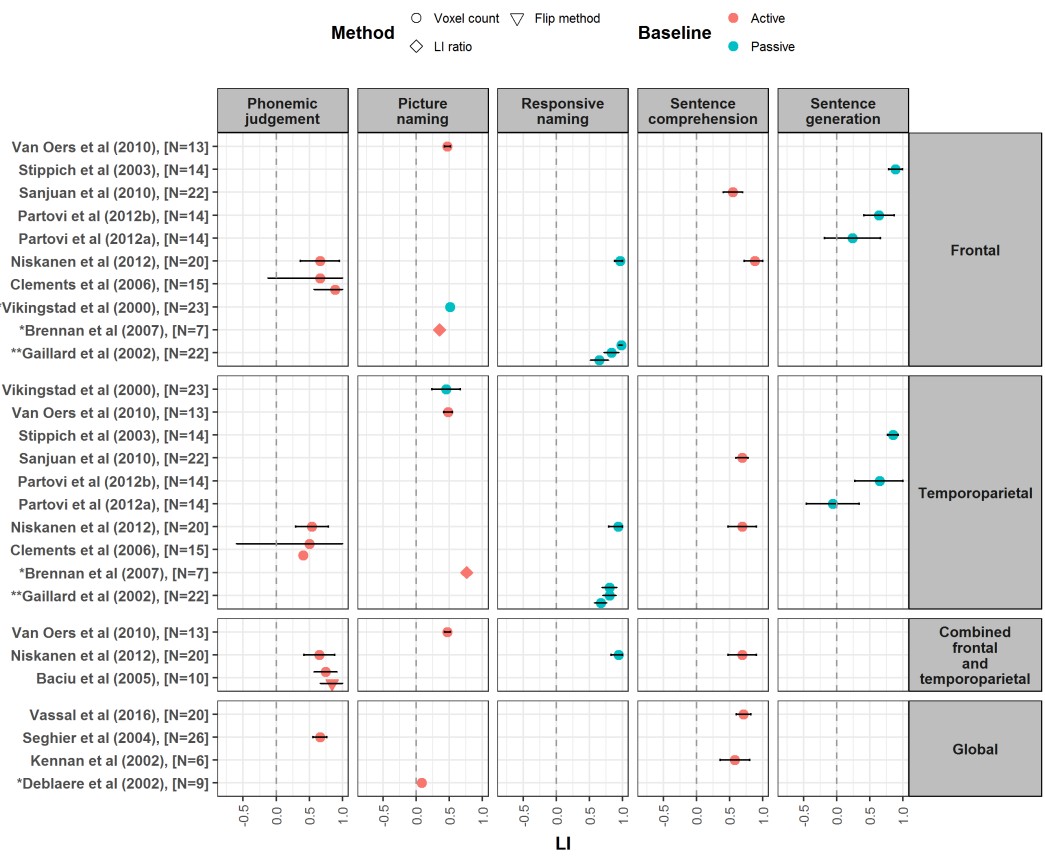

**Figure 3** **Forest plot of mean LI values for phonemic judgement, naming, sentence comprehension and sentence generation tasks.** Forest plot shows mean LI values for different language tasks reported from studies meeting our criteria. Plot is divided up according to region of interest used for LI calculation (frontal, temporoparietal, combined frontal and temporoparietal and global). Error bars represent 95% confidence limits. Colour of symbol indicates type of baseline task used (active or passive), and shape of symbol indicates method of LI calculation (see key). *Papers did not report a measure of spread for LI values, so confidence interval is not shown. **LI values reported by this paper are given at different thresholds: $t = 5$ (Top), $t = 4$ (Middle) and $t = 3$ (Bottom).

fluency), belonging to a particular semantic category (semantic fluency), verbs that are semantically associated with a particular noun (verb generation), or words that are antonyms/synonyms (antonym/synonym generation). Any lateralisation induced by this task may thus reflect a mixture of phonological, semantic, word retrieval and speech motor planning/articulation processes (see Table 2). Lateralisation of speech motor processes is a subject of debate (see Table 1), with some considering them left lateralised (*Hickok & Poeppel, 2007*), but others bilateral (*Poeppel, 2014*; *Price, 2012b*).

### LI strength, reproducibility and robustness

Across the papers reviewed here, verbal fluency tasks are consistently reported as yielding the strongest laterality when compared to other receptive and expressive tasks within

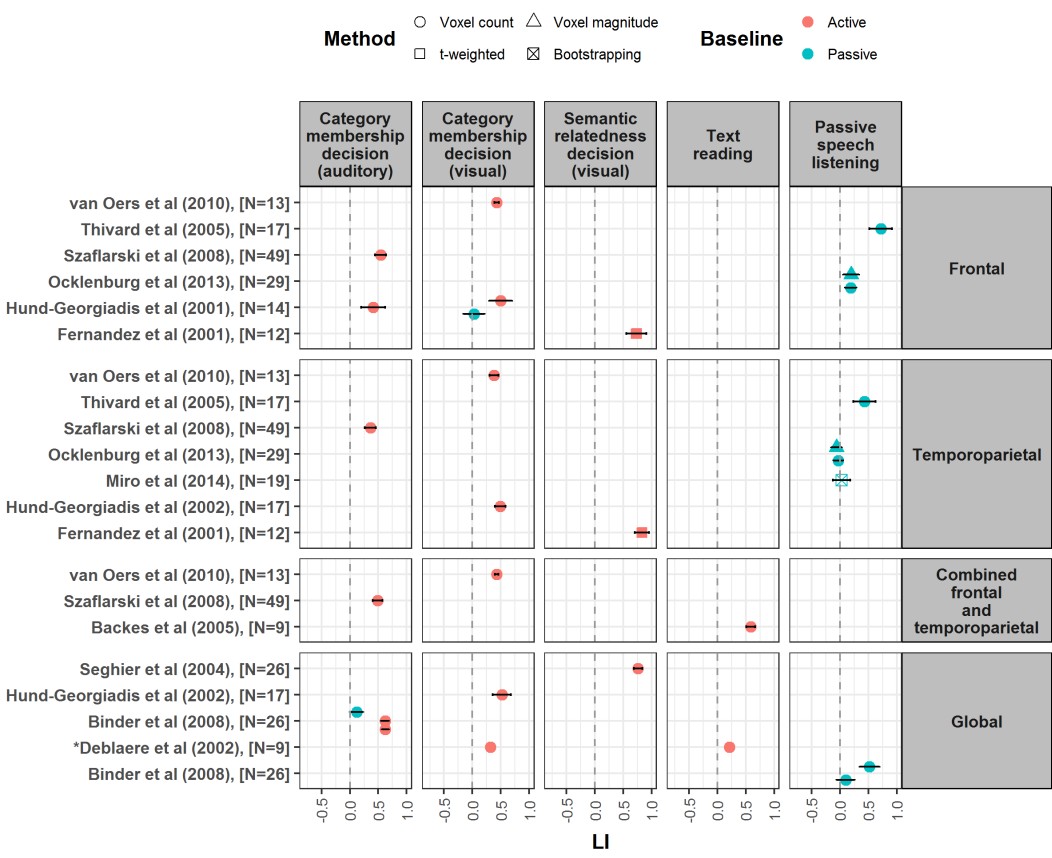

**Figure 4  Forest plot of mean LI values for semantic decision, text reading and speech listening tasks.**
Forest plot shows mean LI values for different language tasks reported from studies meeting our criteria.
Plot is divided up according to region of interest used for LI calculation (frontal, temporoparietal, combined frontal and temporoparietal and global). Error bars represent 95% confidence limits. Colour of symbol indicates type of baseline task used (active or passive), and shape of symbol indicates method of LI calculation (see key). *Papers did not report a measure of spread for LI values, so confidence interval is not shown.

studies (*Baciu et al., 2005*; *Deblaere et al., 2002*; *Van der Haegen, Cai & Brysbaert, 2012*; *Dodoo-Schittko et al., 2012*; *Harrington, Buonocore & Farias, 2006*; *Jensen-Kondering et al., 2012*; *Niskanen et al., 2012*; *Ocklenburg, Hugdahl & Westerhausen, 2013*; *Ramsey et al., 2001*; *Vikingstad et al., 2000*; *Zaca, Jarso & Pillai, 2013*). Studies included in the forest plots produced here report a wide spread of LI values for verbal fluency tasks, ranging from 0.05 to 0.94 (see Fig. 2).

A number of studies have compared the use of different verbal fluency paradigms. When a frontal region of interest (ROI) is used, semantic fluency is reported as less strongly lateralising than verb generation or phonemic fluency (*Kleinhans et al., 2008*; *Ruff et al., 2008*; *Sanjuan et al., 2010a*). Conversely, two studies using combined frontal and temporoparietal ROIs reported no differences in their strength of laterality (*Rutten et al., 2002*; *Tailby et al., 2014*). This suggests that these tasks may differ in the extent of lateralisation they induce across different language areas (see following section, effect of

Bradshaw et al. (2017), *PeerJ*, DOI 10.7717/peerj.3929

**Table 2  Language processes engaged by different language and baseline tasks.** Table shows the main language tasks (top left quadrant) and baseline tasks (bottom left quadrant) identified as being widely used in laterality research. For each type of task, the number of studies (*N*) within our search selection using this task is given, as well as one characterisation of the different language processes (middle column) and domain general processes (right column) they engage.

| Task | N studies | Language processes | | | | | | Domain general processes | | |
|---|---|---|---|---|---|---|---|---|---|---|
| | | Speech motor planning/articulation | Phonological access | Orthographical processing | Semantics | Word retrieval | Syntax | Working memory | Motor processing | Auditory processing |
| **Language tasks** | | | | | | | | | | |
| Verbal fluency | 53 | ✔ | ✔ | (✔) | (✔) | ✔ | | ✔ | | |
| Sentence generation | 5 | ✔ | ✔ | | ✔ | ✔ | ✔ | ✔ | | |
| Passive speech listening | 7 | | ✔ | | ✔ | | ✔ | (✔) | | ✔ |
| Text reading | 2 | ✔ | ✔ | ✔ | ✔ | | ✔ | (✔) | | |
| Phonemic decision | 8 | (✔) | ✔ | (✔) | (✔) | (✔) | | ✔ | (✔) | |
| Semantic decision | 20 | | (✔) | (✔) | ✔ | (✔) | | ✔ | (✔) | |
| Sentence comprehension | 8 | | ✔ | (✔) | ✔ | | ✔ | ✔ | | (✔) |
| Naming | 9 | ✔ | ✔ | | ✔ | ✔ | | | | |
| **Baseline tasks** | | | | | | | | | | |
| Rest | 37 | | | | (✔) | | | | | |
| Perceptual decision (non-linguistic) | 25 | | | | | | | ✔ | (✔) | |
| Finger tapping | 3 | | | | | | | | ✔ | |
| Non-word/word repetition | 8 | ✔ | ✔ | | | | | | | |
| Recite months of the year/count sequence | 2 | ✔ | ✔ | | (✔) | ✔ | | | | |
| Tone listening | 3 | | | | | | | | | ✔ |
| Backward speech listening | 2 | | | | | | | | | ✔ |
| Nonsense text reading | 2 | ✔ | ✔ | ✔ | | | | | | |
| Spatial position naming | 1 | ✔ | ✔ | | ✔ | ✔ | | | | |

**Notes.**

✔, engaged; (✔), sometimes engaged (e.g., depending on task demands, modality of stimuli, occurrence of automatic linguistic processing).

region of interest). Interestingly, multiple studies report that LIs from generation tasks can vary substantially depending on methodological choices made when calculating laterality, such as the threshold chosen (*Dodoo-Schittko et al., 2012*), the use of normalisation, smoothing and clustering techniques (*Baciu et al., 2005*), and the activity measure used (*Harrington, Buonocore & Farias, 2006*).

### Effect of region of interest

Verbal fluency tasks tend to induce the strongest laterality in frontal ROIs (*Gaillard et al., 2003*; *Niskanen et al., 2012*; *Ocklenburg, Hugdahl & Westerhausen, 2013*; *Partovi et al., 2012a*; *Partovi et al., 2012b*; *Propper et al., 2010*; *Szaflarski et al., 2008*; *Vernooij et al., 2007*; *Vikingstad et al., 2000*; *Vingerhoets et al., 2013*; *Zaca, Jarso & Pillai, 2013*). Although they can induce strong laterality in temporoparietal ROIs (*Harrington, Buonocore & Farias, 2006*; *Jensen-Kondering et al., 2012*; *Stippich et al., 2003*), this may not be significantly greater than other tasks within this ROI (*Zaca, Jarso & Pillai, 2013*). However, this may depend on the particular fluency task used; *Jensen-Kondering et al. (2012)* reported that while phonemic fluency and verb generation yielded the strongest lateralisation for a frontal ROI, the strongest laterality for a temporoparietal ROI was seen with semantic fluency, consistent with a role for such areas in semantic cognition.

### Effect of baseline task

Although the majority of studies using verbal fluency tasks employed a passive baseline task such as fixation, a number used active baselines such as finger tapping or silent word repetition (see Fig. 2 and Table 2). *Dodoo-Schittko et al. (2012)* reported that an active baseline task which required subvocal manipulation of the order of syllables within a pseudoword yielded significantly stronger laterality for a verb generation task compared to the use of a passive resting baseline. This is consistent with the idea of subtraction of bilateral activity related to speech-motor planning (*Poeppel, 2014*; *Price, 2012b*).

## Sentence generation

Sentence generation requires participants to generate sentences to describe presented pictures. These sentences may either be pre-defined and learnt prior to scanning, or generated during the scan itself. Relative to word generation, additional syntactic and semantic-integration processes are involved in the construction of a sentence (see Table 2). These are argued to be left lateralised by multiple models (*Peelle, 2012a*; *Poeppel, 2014*). *Poeppel*'s (*2014*) COM-PRE hypothesis makes a distinction between bilateral processing within input and output interfaces (e.g., auditory perception and speech production), and left dominant processing of combinatorics and composition (COM) or linguistically-based predictions (PRE). Similarly, *Peelle (2012)* predicts that while unconnected language (e.g., single words) is processed bilaterally, processing of connected language that requires more complex linguistic operations is left lateralised. Thus, these models might predict stronger laterality for sentence generation over word generation paradigms, due to the additional sentential processing demands.

### LI strength, reliability and robustness

Mean LIs reported from sentence generation studies are illustrated in Fig. 3. High mean LIs of between 0.74 and 0.89 have been reported for sentence generation, both when sentences are pre-learnt prior to scanning (e.g., *Stippich et al., 2003*), and when they are actively generated during the scan (e.g., *Tzourio-Mazoyer et al., 2016*). However, other studies have reported more modest laterality estimates of between 0.48 and 0.65, again with both variants of the task (*Mazoyer et al., 2014*; *Partovi et al., 2012a*; *Partovi et al., 2012b*). Thus, it does not appear to be the case that strength of laterality differs according to whether sentences are generated spontaneously during the scanning session, or learnt prior to scanning. Two studies within our search measured laterality for semantic fluency and sentence generation within the same participants (*Partovi et al., 2012a*; *Stippich et al., 2003*); however, these studies reported differences in strength of laterality between the tasks in different directions. Further, they used the version of the sentence generation task in which sentences are learnt prior to scanning, so task demands were not well matched. Thus, there is currently insufficient data with these tasks to evaluate predictions of stronger laterality for sentence over word processing. *Partovi et al. (2012a)* report good reproducibility for sentence generation, however their analysis simply looked at significant differences in group means over repeated testing, and not at reproducibility of individual participant's LIs.

### Effect of ROI

There is mixed evidence as to whether sentence generation yields differences in the laterality measured from frontal versus temporoparietal ROIs. While *Partovi et al. (2012a)*, *Partovi et al. (2012b)* and *Stippich et al. (2003)* reported equivalent strength of laterality across both, *Tzourio-Mazoyer et al. (2016)* found significantly stronger laterality in frontal than temporal areas. Interestingly, in contrast to the exclusively right handed samples of the former studies, this latter study used a mixed handedness sample, with an overrepresentation of left handers. This suggests the possibility that greater regional heterogeneity may characterise the atypical profiles of language lateralisation that are more often found within atypical handedness samples.

### Effect of baseline

When sentences are learnt prior to scanning, studies generally employ simple cross-fixation or rest as a baseline (*Partovi et al., 2012a*; *Partovi et al., 2012b*; *Stippich et al., 2003*). Conversely, the two studies using spontaneous generation of sentences during the scanning session used an active linguistic baseline, in which participants covertly generated the months of the year (*Mazoyer et al., 2014*, *Tzourio-Mazoyer et al., 2016*). A comparison of these baselines in terms of the language processes isolated by each contrast is given in Table 3. As can be seen, the active baseline subtracts out activity related to speech motor planning to leave those processes specific to the construction of novel sentences, such as syntactic and lexico-semantic processing; conversely, the contrast with rest results in poor isolation of such language processes. This highlights the need to consider carefully the functions one wishes to isolate when choosing a suitable baseline, and the implications this will have for interpretation of measured laterality in relation to linguistic processes.

**Table 3  Language processes isolated by subtraction of different baseline tasks from sentence generation.** The table shows comparison of passive and active baseline tasks as used for subtraction with sentence generation, in terms of the language and domain-general processes engaged by each paradigm and isolated by the subtraction contrast.

| Contrast | Speech motor planning/ articulation | Phonological access | Orthographical processing | Semantics | Word retrieval | Syntax | Working memory | Motor processing | Auditory processing |
|---|---|---|---|---|---|---|---|---|---|
| Task: sentence generation | ✔ | ✔ | | ✔ | ✔ | ✔ | ✔ | | |
| Baseline: rest | | | | (✔) | | | | | |
| Sentence generation vs rest | ✔ | ✔ | | (✔) | ✔ | ✔ | ✔ | | |
| Task: sentence generation | ✔ | ✔ | | ✔ | ✔ | ✔ | ✔ | | |
| Baseline: recite months | ✔ | ✔ | | (✔) | (✔) | | | | |
| Sentence generation vs recite months | | | | (✔) | (✔) | ✔ | ✔ | | |

Notes.

✔, engaged; (✔), sometimes engaged (e.g., depending on task demands, modality of stimuli, occurrence of automatic linguistic processing).

## Passive speech listening

Passive speech listening as a language paradigm appears to yield more variable laterality estimates, perhaps reflective of the wide variety of language processes that it can engage (see Table 2). Lower-level acoustic processing of speech sensory input is predicted to be bilateral by multiple models (*Hickok & Poeppel, 2007*; *Peelle, 2012*; *Poeppel, 2014*; *Price, 2012b*). However, there are discrepancies in the extent of lateralisation assumed for mapping of sound to meaning, considered bilateral by *Hickok & Poeppel (2007)* but left lateralised by other authors (*Peelle, 2012*; *Poeppel, 2014*; *Price, 2012b*) owing to the need to process meaning at a sentential level. Thus, depending on the baseline subtraction used, different levels of processing may be isolated to result in variable levels of laterality.

### LI strength, reliability and robustness

The majority of studies using passive listening tasks reported very weak average LIs (see Fig. 4 for mean LI values), indicating bilateral activation; indeed, passive listening is often the most weakly lateralising task when compared to others (*Binder et al., 2008*; *Harrington, Buonocore & Farias, 2006*; *Miro et al., 2014*; *Ocklenburg, Hugdahl & Westerhausen, 2013*; *Tzourio-Mazoyer et al., 2015*). A notable exception to this was presented by *Thivard et al. (2005)* who reported a mean laterality index of 0.72 within a frontal ROI for a passive story listening task, stronger than that seen in this ROI for a semantic fluency task (0.51). We also note that high test–retest correlations have been reported for speech listening (*Razafimandimby et al., 2007*).

### Effect of ROI

Studies are inconsistent as to whether stronger laterality is found for speech listening within a frontal or a temporoparietal ROI. *Harrington, Buonocore & Farias (2006)* reported that temporoparietal LIs were stronger and more reliable than frontal LIs, whereas other studies have reported weaker and more variable LIs for a temporal compared to a frontal ROI (*Miro et al., 2014*; *Ocklenburg, Hugdahl & Westerhausen, 2013*; *Thivard et al., 2005*). In general, posterior language areas appear to be poorly lateralised for receptive speech listening tasks, although this may depend on the baseline task employed (see paragraph below).

### Effect of baseline task

The varying levels of asymmetry reported for speech listening tasks may in part be attributable to the baseline used by different studies. *Binder et al. (2008)* found that changing the baseline from rest to tone listening raised the average LI for word listening from 0.1 to 0.52. In this regard it is interesting that the two studies reporting near-zero average LI values for speech listening employed rest as a baseline task (*Miro et al., 2014*; *Ocklenburg, Hugdahl & Westerhausen, 2013*). Conversely, *Thivard et al. (2005)* and *Harrington, Buonocore & Farias (2006)* both used backwards speech listening as a baseline and reported stronger laterality measurements for speech listening. This effect of baseline is consistent with the idea of bilateral early auditory processing that must be subtracted out by a non-linguistic auditory stimulus in order to reveal asymmetry for higher-level 'central language processes' (*Peelle, 2012*; *Poeppel, 2014*; *Price, 2012*).

## Text reading

Reading text or narrative requires decoding of orthography into phonological representations, semantic and syntactic processing of the decoded sentence, and binding within and across sentences to arrive at an overall understanding of text meaning (see Table 2). Visual word form processing is considered to rely on a lateralised ventral occipito-temporal region, although this may not reflect a left specialisation for orthography *per se* (*Price, 2012b*).

### LI strength and effect of ROI

Our search identified only two papers investigating lateralisation of text reading. Both studies used the same covert (silent) text reading task with a baseline of covert reading of text composed of pronounceable non-words. *Backes et al. (2005)* reported moderately strong laterality (LI = 0.59) using a combined frontal-temporoparietal ROI, whereas (*Deblaere et al., 2002*) reported weak laterality (LI = 0.21) using a global LI (see Fig. 4). This supports the hypothesis that global LIs are generally weaker than regional LIs.

## Phonemic judgement

Phonemic judgement tasks require a decision relating to phonological structure; most commonly, a rhyme judgement. This task relies on mapping of acoustic or visual input onto phonological units such as phonemes and syllables, a process known as decoding. The precise nature of these stored phonological codes remains a debate; according to theorists in the tradition of the motor theory of speech perception (e.g., *Liberman & Mattingly, 1985*), these are represented as speech motor gestures in left premotor cortex. *Hickok & Poeppel (2007)* argue that while the phonological codes themselves are bilaterally represented, the process of their mapping onto articulatory motor representations relies on a left lateralised dorsal stream. Conversely, other models propose that such processing of single words is a less strongly lateralised process (*Peelle, 2012*).

### LI strength, reliability and robustness

Phonemic judgement tasks yield relatively strong laterality, with reported LI values ranging from 0.41 to 0.84 (see Fig. 3). However, when compared to other tasks, phonemic judgement

is often reported as more weakly lateralising (*Baciu et al., 2005*; *Niskanen et al., 2012*; *Seghier et al., 2004*). Phonemic judgement may be superior to other tasks however in terms of robustness and reproducibility. *Morrison et al. (2016)* reported that a rhyming decision task demonstrated greater reliability than a word generation task, yielding reproducible dominance classifications in 100% of participants, and average test-retest correlations for LI values of 0.9 and above. Furthermore, such reproducibility of LIs obtained with rhyming decision was more robust against changes in the activity measure used for LI calculation.

### Effect of ROI

Rhyming decision tasks yield particularly strong laterality when a frontal or a combined frontal-temporoparietal ROI is used (*Baciu et al., 2005*; *Clements et al., 2006*; *Cousin et al., 2007*; *Niskanen et al., 2012*). For example, *Cousin et al. (2007)* identified a particularly strong leftward asymmetry for the inferior frontal gyrus during rhyme detection. Thus, frontal ROIs may be optimal for yielding the strongest laterality with this task.

### Effect of baseline task

All studies within our search using phonemic judgement were found to employ an active perceptual decision baseline task on either non-linguistic material (e.g., line orientation matching) or nonsense words or characters (e.g., nonsense word font matching). This subtracts out non-linguistic working memory processes (see Table 2), as well as basic visual processing.

## Semantic decision

Semantic decision tasks require a judgement about a word's semantic content or about the semantic relationship between a pair of words, such as whether two words belong to the same category. Such conceptual knowledge is proposed to rely on a distributed processing network, with different brain areas each contributing to different aspects of an item's representation (*Warrington & McCarthy, 1983*; *Warrington & McCarthy, 1987*; *Warrington & Shallice, 1984*). In addition to this distributed network, *Patterson, Nestor & Rogers (2007)* have argued for the existence of a 'semantic hub' within the bilateral anterior temporal lobes that integrates the distributed modality-specific representations into one amodal representation. However, a recent meta-analysis of functional imaging studies by *Rice, Ralph & Hoffman (2015)* suggested that while conceptual knowledge does appear to be represented bilaterally in the anterior temporal lobes, left lateralised activity was more likely when semantic content was accessed linguistically. This is in contrast to the predictions of *Hickok & Poeppel*'s (*2007*) model of language in which access to lexico-semantics from speech processing (via the ventral stream) is considered as a bilateral process.

### LI strength, reproducibility and robustness

The strength of laterality reported for semantic decision tasks is quite variable, ranging from near-zero to around 0.8 (see Fig. 4 for mean LI values). This may depend on the type of semantic decision required. Tasks which require judgement of the semantic relatedness of two words appear to yield relatively strong laterality, ranging from 0.59 to 0.84 (*Bethmann et al., 2007*; *Fernandez et al., 2001*; *Häberling, Steinemann & Corballis,*

*2016*; *Seghier et al., 2004*). In contrast, category membership tasks with single words appear to give much lower LIs, ranging from 0.03 to 0.52 (*Deblaere et al., 2002*; *Hund-Georgiadis et al., 2002*; *Hund-Georgiadis, Lex & von Cramon, 2001*; *Ramsey et al., 2001*; *Van Oers et al., 2010*). This suggests that it may be the process of integrating and comparing across semantic representations for different concepts that is strongly lateralising; conversely a simple lexical look-up to determine the category membership of a single concept may not be strongly lateralised.

*Jansen et al. (2006)* reported very low reproducibility for synonym decision LIs across a range of different LI calculation methods within Broca's area; much higher reproducibility was found however when a temporoparietal ROI was used. Conversely, *Harrington, Buonocore & Farias (2006)* reported high test-retest correlations for an abstract/concrete semantic decision task across both frontal (IFG) and temporoparietal ROIs. This discrepancy between the two studies could be due to the differences in the tasks they used.

### Effect of ROI

The majority of studies report no significant differences in the magnitude of laterality found within temporoparietal and frontal ROIs for semantic decision tasks (*Bethmann et al., 2007*; *Häberling, Steinemann & Corballis, 2016*; *Harrington, Buonocore & Farias, 2006*; *Hund-Georgiadis et al., 2002*; *Ramsey et al., 2001*; *Van Oers et al., 2010*). Some studies have reported differences across ROIs, however these can be in opposite directions (*Fernandez et al., 2001*; *Szaflarski et al., 2008*). As discussed, some evidence suggests that LIs calculated from temporoparietal ROIs for semantic decision may be more reproducible than those calculated from frontal ROIs (*Jansen et al., 2006*).

### Effect of baseline tasks

Semantic decision laterality is also strongly influenced by the baseline task used. *Binder et al. (2008)*, *Hund-Georgiadis et al. (2002)* and *Hund-Georgiadis, Lex & Von Cramon (2001)* manipulated the baseline and found that the use of an active perceptual decision task as opposed to passive rest yielded a large increase in the strength and consistency of semantic decision laterality. *Binder et al. (2008)* argued that resting baselines are unsuitable for subtraction with semantic decision, since they allow for the activation of conceptual language representations as the participant 'day dreams' and engages in 'inner speech'. An active perceptual decision baseline interrupts such ongoing conceptual processing and engages the same executive and attentional processes as the language paradigm. This subtraction is shown in Table 4, which illustrates the better isolation of semantic processes that this baseline provides compared to the contrast with rest. Baseline tasks that engage linguistic processing themselves may result in reduced laterality; for example, *Deblaere et al. (2002)* suggested their finding of weak laterality for semantic decision may have been due to a vowel decision baseline task. However, it should be noted that *Binder et al. (2008)* reported identical laterality strength (a mean LI of 0.62) for semantic decision using either a baseline of tone decision or phoneme decision. Overall, this evidence suggests that baseline

Table 4  **Language processes isolated by subtraction of different baseline tasks from semantic decision.** Table shows comparison of active and passive baseline tasks as used for subtraction with semantic decision, in terms of the language and domain-general processes engaged by each paradigm and isolated by the subtraction contrast.

| Contrast | Speech motor planning/ articulation | Phonological access | Orthographical processing | Semantics | Word retrieval | Syntax | Working memory | Motor processing | Auditory processing |
|---|---|---|---|---|---|---|---|---|---|
| Task: semantic decision | | (✔) | (✔) | ✔ | (✔) | | ✔ | (✔) | |
| Baseline: rest | | | | (✔) | | | | | |
| Semantic decision vs rest | | (✔) | (✔) | (✔) | (✔) | | ✔ | (✔) | |
| Task: semantic decision | | (✔) | (✔) | ✔ | (✔) | | ✔ | (✔) | |
| Baseline: perceptual decision | | | | | | | ✔ | (✔) | |
| Semantic vs perceptual decision | | (✔) | (✔) | ✔ | (✔) | | | | |

**Notes.**
✔, engaged; (✔), sometimes engaged (e.g., depending on task demands, modality of stimuli, occurrence of automatic linguistic processing).

tasks used for semantic decision must be active, sufficiently engaging and challenging so as to prevent 'day-dreaming', and ideally involve material from a non-linguistic domain e.g., symbols or tones.

## Sentence comprehension

Sentence comprehension tasks require some judgement about the content of a spoken or written sentence. Syntactic and semantic processing are often confounded (see Table 2); for example, the task may require participants to decide if two sentences with different grammatical constructions have the same meaning. However, they are noteworthy among other tasks in the extent of their syntactic processing requirements. Laterality of syntax has been a subject of debate, with some authors arguing for a bilateral involvement in syntax (e.g., *Hund-Georgiadis, Lex & Von Cramon, 2001*), but others arguing for a left dominance (*Friederici, 2011*; *Tyler et al., 2011*; *Wright, Stamatakis & Tyler, 2012*). At a more general level, multiple models would predict left lateralisation for the sentence-level processing engaged by this task, without making specific claims about lateralisation of syntactic processing *per se* (e.g., *Peelle, 2012*; *Poeppel, 2014*; *Price, 2012*).

### LI strength, reliability and robustness

Multiple studies report strong laterality for sentence comprehension tasks (*Harrington, Buonocore & Farias, 2006*; *Jensen-Kondering et al., 2012*; *Kennan et al., 2002*; *Niskanen et al., 2012*; *Sanjuan et al., 2010b*; *Vassal et al., 2016*), with LI values ranging from 0.55 to 0.88 (see Fig. 3). Studies which compare sentence comprehension laterality measures to those of other tasks suggest that it can outperform semantic decision, phoneme decision, story listening and naming tasks in terms of the strength of laterality, although this can depend on the ROI (*Harrington, Buonocore & Farias, 2006*; *Niskanen et al., 2012*).

### Effect of ROI

Evidence appears inconsistent as to the effect of ROI on the laterality obtained with sentence comprehension tasks. Studies have reported both stronger laterality for frontal than temporoparietal ROIs (*Jensen-Kondering et al., 2012*; *Niskanen et al., 2012*) and vice versa (*Harrington, Buonocore & Farias, 2006*; *Sanjuan et al., 2010b*). In terms of reliability

of lateralisation, *Harrington, Buonocore & Farias (2006)* reported very high reproducibility for a visual sentence comprehension task across both frontal and temporoparietal ROIs, with test-retest correlations above 0.9. In contrast, auditory sentence comprehension yielded more reliable lateralisation within a temporoparietal than a frontal ROI. Modality of the stimuli may thus affect which ROI is optimal.

### Effect of baseline task

Generally, active baselines are employed for sentence comprehension tasks. A notable exception is seen in *Harrington, Buonocore & Farias (2006)* who used passive listening to backwards speech as a baseline for auditory sentence comprehension. This passive baseline may explain the weaker laterality they reported as compared to other studies (mean LI of around 0.45). Interestingly, *Sanjuan et al. (2010a)*; *Sanjuan et al. (2010b)* who also reported a relatively low level of laterality compared to other studies used phoneme decision as a baseline. As previously discussed in relation to a study by *Deblaere et al. (2002)* using semantic decision, it is possible that the use of such a baseline with high linguistic processing demands may lower the strength of the laterality seen.

## Naming

Naming tasks require the generation of the name of an item in response to either a visual (pictorial) or verbal description. According to *Hillis*' (*2007*) model of naming, picture naming involves three major levels of processing; a semantic level in which amodal general and specific semantic information is accessed from a structural description of an object; a lemma level which involves the defining features of an object at a more abstract level (e.g., what makes a sheep a sheep); and a phonological/orthographical level, in which the phonological and orthographical representations associated with that concept are accessed. Thus, naming tasks have the potential to engage multiple key components of the language network (see Table 2).

### LI strength, reliability and robustness

Studies using naming as a language activation task report a wide variety of LI values, ranging from 0.08 to 0.96 (see Fig. 3). This can partly be explained by variation in the nature of the naming task used. Zero to moderate laterality has been reported for picture naming tasks, which are often the least lateralising when compared to other tasks (*Deblaere et al., 2002*; *Harrington, Buonocore & Farias, 2006*; *Jansen et al., 2006*; *Van Oers et al., 2010*; *Vikingstad et al., 2000*). However, the lateralising ability of naming tasks can be increased by the addition of sentence comprehension demands. A naming from written or auditory description task known as 'responsive naming' requires comprehension of a question describing an object in order to generate the required name. This has been reported to yield strong laterality across both frontal and temporal ROIs, in the range of 0.65 to 0.96 (*Gaillard et al., 2002*; *Niskanen et al., 2012*). This increase in laterality with the addition of sentence-level processing is consistent with models of language which predict an increase in laterality for connected versus unconnected language i.e., structured sentences versus single words (*Peelle, 2012*; *Poeppel, 2014*).

Picture naming also shows poor reliability in laterality measurement. *Jansen et al. (2006)* reported that picture naming did not determine dominance reproducibly in about a third of participants. *Rutten et al. (2002)* similarly reported a failure to find significant test-retest correlations for naming LIs. Significant test-retest correlations were reported by *Harrington, Buonocore & Farias (2006)* for a picture naming task at around the same level as those seen for semantic decision; however, reproducibility of naming laterality was lower than that seen for verb generation and sentence comprehension.

### Effect of ROI

Naming tasks do not appear to favour one ROI over another in laterality measurement. We found two studies which reported differences in the laterality measured from frontal and temporoparietal ROIs, however this difference was in opposite directions (*Harrington, Buonocore & Farias, 2006*; *Brennan et al., 2007*). The majority of studies instead report highly similar strength of laterality across frontal and temporoparietal ROIs (*Gaillard et al., 2002*; *Niskanen et al., 2012*; *Van Oers et al., 2010*; *Vikingstad et al., 2000*). Furthermore, *Rutten et al. (2002)* reported similar levels of reproducibility of naming LIs for both regions.

### Effect of baseline task

The baselines used with picture naming provide interesting evidence on the processes underlying its laterality. *Deblaere et al. (2002)* reported near zero laterality for picture naming using a baseline which required participants to name the position of the intersection of four lines (e.g., up, down, left, right). This task involves engagement in similar semantic and word retrieval processes (see Table 2), which may explain the weak laterality measured. However, *Brennan et al. (2007)* reported strong laterality for a picture naming task with a number counting baseline. This would subtract out speech production and word retrieval processes for an automated speech sequence, predicted to involve bilateral activity (*Price, 2012b*; *Poeppel, 2014*). This subtraction would thus isolate spontaneous non-automated retrieval and word generation processes which may engage left hemisphere language systems, increasing measured laterality.

## Combined task analysis

Combined task analysis (CTA) involves the calculation of LIs from contrast images generated by combining scans across multiple language tasks. This method identifies commonalities between tasks' activity patterns in order to isolate the 'core' language network, and exclude task-specific, non-linguistic activity caused by differences in task design that may influence the LI value. In this way, CTA can represent a theoretical alternative to baseline tasks to subtract domain-general activity, assuming that different tasks involve different patterns of non-linguistically relevant activity. Indeed, there is evidence that CTA results in higher and more reliable and robust estimates of laterality for language (*Dodoo-Schittko et al., 2012*; *Harrington, Buonocore & Farias, 2006*; *Jansen et al., 2006*; *Niskanen et al., 2012*; *Ramsey et al., 2001*; *Rutten et al., 2002*; *Sommer et al., 2003*; *Van Rijn et al., 2008*).

Nevertheless, the theoretical assumptions which motivate CTA can be questioned. CTA assumes that variability in laterality for different tasks should be ascribed to non-linguistic processes or viewed simply as measurement error, rather than reflecting the underlying nature of hemispheric organisation for language (e.g., *Ramsey et al., 2001*). Such a theoretical stance ignores the possibility of multidimensional lateralisation across different language processes. Indeed, recent fMRI studies have reported cases of dissociated laterality for different language functions within individuals (*Van der Haegen, Cai & Brysbaert, 2012*; *Häberling, Steinemann & Corballis, 2016*; *Vikingstad et al., 2000*); see *Bradshaw, Bishop & Woodhead, 2017* for a review), corroborating those early clinical reports of a 'division of labour' across the hemispheres in patients with bilateral language representation (*Rasmussen & Milner, 1975*). It is not yet known whether such crossed language dominance has significant functional implications for language abilities. *Bishop (2013)* speculated that having expressive and receptive language functions in opposite hemispheres may make one more vulnerable to development of language disorders or impairments.

Such differences in dominance between language tasks would be lost in a CTA, since combining scans across paradigms would result in few areas of common activation and thus a loss of these tasks differences. Instead, it will be necessary to design fMRI protocols that probe the within-subject variation in language lateralisation across a range of tasks, while controlling for non-linguistic confounds. Conversely, the strong, reliable and robust LI values provided by CTA would be more useful in cases where a clear categorical decision on an individual's language lateralisation is required.

## SUMMARY AND CONCLUSIONS

This review has highlighted the high level of variation and inconsistency in the strength and reliability of laterality measured using different language tasks. As per our hypotheses, some of this variability in laterality is related to parameters such as the region of interest and baseline task, which can have task-specific effects. In general, however, the current state of the literature is such that it is difficult to draw clear conclusions that can be used to guide task selection. This review highlights the need for more research that systematically compares laterality across different tasks in within-subject designs, with rigorous matching of non-linguistic aspects of task design.

The current review of the literature does suggest however some practical recommendations that can be used to guide task design. Extensive use of verbal fluency is clearly warranted given the robustness of its lateralisation; however, the common employment of passive baselines should be replaced with active baselines carefully chosen so as to isolate the language process of interest whilst controlling for all other processes. Comparison of word generation with sentence generation offers the opportunity to test predictions of models that assume stronger laterality for processing of connected sentences over single words (*Peelle, 2012*; *Poeppel, 2014*). Future studies should try to more closely match task demands of word generation and sentence generation tasks, in order to systematically compare their strength of laterality within subjects. For example, one could use the same stimuli for each task such as names of different categories, with a cue

indicating whether the subject should generate instances of these categories (semantic fluency) or generate a sentence taking an instance of this category as its head noun.

Evidence on semantic decision paradigms suggests that stronger laterality can be observed when tasks require integration across the semantic content of different concepts (e.g., semantic relatedness decision), rather than simple category membership decision on single words. Similarly, where naming tasks are used, evidence suggests that naming from description yields more robust laterality measurement than naming from pictures; that is, an additional sentence comprehension component appears to improve the lateralising power of this task. Sentence comprehension tasks themselves appear to yield strong laterality; however, more work is needed to develop such tasks in order to attempt to disentangle semantic and syntactic components. Indeed, this review highlights a distinct lack of tasks in language laterality research that aim to primarily engage syntactic processing, reflected in the lack of consensus over the strength of its hemispheric specialisation. Further work is needed to design and validate tasks that isolate syntactic processing for laterality measurement. One possibility might be offered by tasks involving judgements on 'jabberwocky' sentences (e.g., *Fedorenko, Nieto-Castanon & Kanwisher, 2012*) in which content words are replaced by non-words (thus preserving syntactic structure but removing semantic content).

More work is needed to investigate the potential significance of variability in laterality across different language functions, both within individuals and at a group level. Growing appreciation of the potential significance of cases of dissociated dominance, both in clinical and healthy samples, should encourage the field to move away from the use of single tasks and single ROIs. Instead, research should focus on developing batteries of closely matched tasks that tap a variety of language functions to allow systematic comparisons in within-subject studies. This will ultimately allow for more quantitative meta-analyses of such literature, to draw stronger conclusions as to patterns of laterality across different components of the language network.

One way to approach this would be to develop a generic task format in which the participant is always performing the same form of task with the same type of stimuli but with regards to different linguistic parameters. For example, one such format could be a decision task in which one must decide if pairs of word stimuli are 'matching' or 'non-matching'. The parameters that define matching and non-matching pairs can then be varied according to the language process of interest; for example, rhyming versus non-rhyming (phonology), same semantic category or different semantic category (semantics) or same syntactic category or different syntactic category (syntax). These could be interleaved, with a visual cue indicating which decision should be made on the current trial. In this way, a generic task format would remove non-linguistic differences in task design that can confound interpretation of differences in laterality. This type of approach has been embraced by Price and colleagues in the development of a battery of tasks devised for a within-subject, fully balanced factorial design, with tasks corresponding to all possible combinations of levels of factors relating to experimental design aspects (e.g., stimulus modality, linguistic content, form of response). This has been used to test contrasts that allow fractionation of different levels of linguistic processing for localization of brain
activity (e.g., *Hope et al., 2014*); future work could implement a similar battery of balanced tasks for lateralisation measurement.

CTA can provide an efficient method of isolating language activity shared across multiple different aspects of language functioning, to allow robust and reliable measurement of laterality of the core language network. However, this methodology appears to be motivated by an implicit assumption pervasive across much laterality research that there is a single core language network which displays a unitary and perfect lateralisation; thus the ability of an fMRI protocol to provide a good measure of language laterality depends on its ability to uniquely engage this language network and to yield a laterality index of close to 1 at the group level. Tasks which yield LIs further from 1 therefore are viewed as inadequate measures of language lateralisation.

We argue that defining the sensitivity of a task to capture the 'true' lateralisation of a language function in terms of the strength of its laterality can be challenged. Such an approach would lead one to reject tasks that yield lower LIs, which may in fact reflect meaningful variation in hemispheric organisation within the language network. For example, naming is a complex linguistic function that requires both receptive and expressive components, with access to both semantics and phonology; however the evidence reviewed here shows it yields low LI values (e.g., *Deblaere et al., 2002*). Is it appropriate to conclude that naming is therefore a 'poor' measure of language network function? Or rather, could this tell us something about the hemispheric organisation of the language functions on which it relies? We would argue that research should be open to the possibility that it may be possible to validly and reliably measure laterality for a language process, and yet still obtain a low LI.

This raises the question of how one should judge laterality paradigms; what metric should one use for judging 'success' in accurately measuring an individual's laterality? This review has highlighted how different methods of laterality measurement can result in variable LI values for an individual across different regions, active language tasks and baseline tasks. For example, in the case of verbal fluency tasks, an individual may show a stronger LI when an active rather than a passive baseline is used (*Dodoo-Schittko et al., 2012*). How should one then decide which of these can be considered to best reflect the 'true' laterality of an individual? In this case, the greater strength of laterality with an active baseline is often taken to indicate that this is a more accurate laterality measurement; however, other metrics such as the reliability of the laterality and in clinical work its predictiveness of post-surgical outcomes may arguably represent better standards for assessing goodness of laterality measurement. In this way, it will be important for the field to consider more deeply the metrics that are used to compare the relative utility of LIs yielded by different paradigms, and to challenge the implicit 'strongest is best' assumption that commonly guides interpretation of task LI values.

### Funding

This work was supported by an Advanced Grant awarded by the European Research Council (project 694189—Cerebral Asymmetry: New directions in Correlates and Etiology —CANDICE). Dorothy Bishop is funded by a programme grant 082498/Z/07/Z from the Wellcome Trust. There was no additional external funding received for this project. The funders had no role in study design, data collection and analysis, decision to publish, or preparation of the manuscript.

### Grant Disclosures

The following grant information was disclosed by the authors:
European Research Council.
Wellcome Trust: 082498/Z/07/Z.

### Competing Interests

Dorothy Bishop is an Academic Editor and Academic Advisor for PeerJ.

### Author Contributions

- Abigail R. Bradshaw performed the experiments, analyzed the data, wrote the paper, prepared figures and/or tables.
- Paul A. Thompson contributed reagents/materials/analysis tools, prepared figures and/or tables.
- Alexander C. Wilson analyzed the data, prepared figures and/or tables.
- Dorothy V.M. Bishop conceived and designed the experiments, reviewed drafts of the paper.
- Zoe V.J. Woodhead reviewed drafts of the paper.

### Data Availability

Bradshaw, Abigail; Thompson, Paul (2017): Forest Plots of LI values for different language tasks. figshare.

https://doi.org/10.6084/m9.figshare.4977950.v3.

GitHub (code): https://github.com/p1981thompson/Candice.

OSF (code and plots):

https://osf.io/7s4hv/.

### Supplemental Information

Supplemental information for this article can be found online at http://dx.doi.org/10.7717/peerj.3929#supplemental-information.

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
