# Peer review of "Measuring language lateralisation with different language tasks: a systematic review"

_PeerJ, doi:10.7717/peerj.3929_

## Round 0.1 · original submission · Minor Revisions

I have now received three reviews on your paper. I thank the reviewers for their work. As you will see, all reviews are thoughtful and very constructive. The three reviewers converge in appreciating your work, and I agree with their assessment.

There are a number of minor issues you might want to assess in a revision. In particular, I think you should try to better outline the hypotheses you start from (see reviewer 3), addressing some of the important theoretical questions raised by reviewer 2.

Thank you for sending your interesting work to PeerJ.

Reviewer 1 ·

Basic reporting

Overall, this is a clearly-written manuscript that thoroughly describes the process of conducting a systematic review. The authors' literature search is thorough and appropriately documented with a Prisma diagram. The parameters for the review are described on osf.io.

The authors note making the R code for the forest plots available at https://osf.io/t24rv/, but when I access this link I get an error - does this need to be made public?

Experimental design

The research question - namely, how different experimental tasks affect language localization - is timely and important. I am not aware of other reviews (certainly not systematic ones) that address this topic. The methods are described in enough detail that the searches could be replicated.

Validity of the findings

The conclusions are well stated and relate to the authors' findings.

Additional comments

Overall I very much enjoyed reading this manuscript, and my lack of substantive comments is due to the excellent job you have done with the manuscript.

I did think that it would be nice to more thoroughly address the various approaches to quantifying lateralization, but i understand the rationale for putting these in a separate paper.

Other minor comments:

-It would be worth having a careful proofread of the references. In my quick glance, I noticed: some journals underlined and italicized (most just italicized); some unusual capitalization (Bmc instead of BMC, Neuroimage rather than NeuroImage); for Hickok & Poeppel (2007) I don't generally see "Opinion" included in the article title; Peelle (2012) is incorrect in the main text (Peele) but correct in the references; some citations have month included (e.g. Warrington & Shallice 1984) but most don't.

-For the Prisma diagram (Figure 1) it would be nice to include reasons for the 8 excluded articles.

-For the figshare link (https://figshare.com/articles/Forrest_Plots_of_LI_values_for_different_language_tasks/4977950), I believe "Forest" plots are mis-spelled, which might lead some people to have difficulty accessing the link. Also, if you want to share the graphs, I would suggest a smaller file size: PNG for raster-based, or perhaps a vector-based format such as .eps.

·

Basic reporting

Well written and generally clear.
See comments under "General comments for the authors".

Experimental design

NA

Validity of the findings

NA

Additional comments

Abstract
Line 32: What does “closely matched tasks” mean? I think the authors mean tasks with baselines that actively control for as many factors as possible, isolating the linguistic process in question. This could be stated more clearly in the abstract (at this first mention of ‘closely matched’).

Introduction
Lines 83-85: “Answering such questions requires optimisation of protocols for LI measurement, to ensure that variability in LIs measured for different language functions cannot be attributed to differences in task sensitivity or measurement error.” I understand what the authors are arguing for here, but is this really even possible? How does one equate apples (e.g. language production) and oranges (e.g. language comprehension)? For instance, how does one define sensitivity without becoming circular (depending upon the LI)?


Results
Line 382: “This subtracts out non-linguistic work memory processes (see Table 3).” Probably worth noting that it subtracts out basic visual processing as well. Again, this highlights the futility of the idea of pure insertion though, as what is essentially a phonemic task, mediated via visual means (orthography), is contrasted with what is essentially a visuospatial form task.

Lines 392-397: “…a recent meta-analysis of functional imaging studies by Rice, Ralph and Hoffman (2015) suggested that while conceptual knowledge does appear to be represented bilaterally in the anterior temporal lobes, left lateralised activity was more likely when semantic content was accessed linguistically. This is in contrast to the predictions of Hickok and Poeppel’s (2007) model of language in which mapping of sound to meaning is considered as a bilateral process.”
Is this really a problem for the Hickock and Poeppel model? Doesn’t it just suggest that linguistic information is left lateralised.

Line 467: A carriage return is missing before, “Effect of ROI”

Line 545: “be” appears to be missing between “may” and “influencing”

Lines 545-546: “In this way, CTA represents a theoretical alternative to baseline tasks to subtract domain-general activity.” This is not really true as the activation analysis for any given task in a CTA still depends on the definition of a baseline.

Lines 557-559: in the references cited here (dissociated language functions within healthy individuals) I would suggest adding (though in the interests of transparency I note that I am an author on one of the papers cited below):
• Tailby, C., Abbott, D. F., & Jackson, G. D. (2017). The diminishing dominance of the dominant hemisphere: Language fMRI in focal epilepsy. NeuroImage: Clinical, 14, 141-150.
• Berl, M. M., Zimmaro, L. A., Khan, O. I., Dustin, I., Ritzl, E., Duke, E. S., ... & Gaillard, W. D. (2014). Characterization of atypical language activation patterns in focal epilepsy. Annals of neurology, 75(1), 33-42.

Line 566: Why would CTA be inappropriate for investigating dissociated language functions within individuals? I don’t follow the logic here – if there really is dissociation (say left Wernicke’s and right Broca’s) why should that not come up across tasks (any less so than the localisation of Wernicke’s and Broca’s in non-dissociated individuals)?


Summary and Conclusions
Line 583: the phrase “active baselines” should be qualified, along the lines of say, “carefully selected active baseline aimed at controlling for all but the language related process of interest.” I don’t think an active baseline per se is the answer the authors are proposing, but rather one that is crafted according to the language laterality question that is being addressed. I think this point is worth belabouring at first appearance in the Discussion.

Lines 588-590: “Future studies should try to more closely match task demands of word generation and sentence generation tasks.” Some further exploration of what kind of form such closer matching might take is called for here, as in some sense this is an (to use fruit as a reference again) apples and oranges question – how does one objectively identify that one has matched word and sentence generation tasks, especially once one has taken into consideration the influence of baseline too?

Lines 611-614: I still do not understand the argument that CTA is somehow inappropriate for dissociated dominance cases, any more so than for non-dissociated individuals (as per comment above). I agree that CTA collapses across inter-task processing particulars, but why this affects dissociated individuals more than non-dissociated individuals eludes me?

Lines 614-616: I agree with the sentiment that “research should focus on developing batteries of closely matched task that tap a variety of language functions to allow systematic comparisons in within-subject studies.” I think, however, that in a manuscript such as this – aimed at providing guidance to future “users” of language lateralisation approaches – that some text should be allocated to sketching out what such an approach might look like. Otherwise statements like the one quoted above just sound like empty rhetoric, rather than being instructive and useful to the reader. Inclusion of a sketch of what such a battery might look like would greatly enhance the utility of the article. As a key take home message from this review article is the importance of ‘closely matched tasks’ I think the authors are beholden to offer some more concrete suggestions here.


Tables
Table 2: ‘Verbal fluency’ should probably have a parenthetic tick in the ‘Orthography’ column, as in the case of letter based fluency (knowing, for instance, that “phlegm” begins with ‘p’ and not ‘f’). Indeed, an alternative name for letter based fluency is Orthographic Lexical Retrieval.
Similarly, ‘Phonemic judgement’ should probably have a tick in the ‘Speech motor planning’ column, given that these kinds of tasks almost invariable require some kind of subvocal rehearsal (arguably also ‘Sentence comprehension’, depending on the length of the sentence).
I happily acknowledge the subjectivity of the table (making categorical judgments about the presence/absence of a given cognition process during a given task), but these seem important constituent elements of these tasks.

More generally, I don’t think that splitting into Table 2 (functions involved in various language tasks) and Table 3 (functions involved in various baseline tasks) is necessarily the most helpful organisation in the context of the goals of the paper. The authors seem to be arguing for the use of better controlled tasks, including better control over differences between active and baseline states in fMRI (e.g. lines 175-176: “Tables 2 and 3 highlight the difficulty in designing a task which isolates a single language function in order to study its laterality”).
For the ‘wishing to be informed’ reader to make best use of this article it would seem ideal to have the same ‘cognitive function’ headings in both Table 2 and Table 3, or even better to have everything combined into one ‘mega-table’ where column one is task (be it active or baseline) and columns 2-n are the purported cognitive functions involved; then one can just scan pairs of rows to find active and baseline states that appear to isolate a given linguistic operation.
For instance, Working Memory is listed as a cognitive function in Table 3 but not in Table 2. I realise I am being a bit literal here but if I use a baseline task for which there is a tick under Working Memory in Table 3, but there is no corresponding Working Memory column in Table 2, what am I as the reader supposed to take from this? Could any deactivation that I see for the language task be due to no working memory load in the active state I have used? A mega-table would be nice in that it would enable the reader to choose a pair of rows, one active and one baseline, and compare where they did or did not have ticks and use this to identify the cognitive operation they are attempting to isolate.
Another option that could prove useful would be to have an example table that goes through the exercise of selecting active and baseline tasks. For instance, using one example language task and a range of different baseline tasks, to illustrate clearly the implications of the decisions made in task design for the observed patterns of activity (e.g. using an example from the text, an active task of sentence generation compared with either (1) a baseline that is passive fixation or (2) a baseline that consists of covert recitation of over-learned sequences like the months of the year).
I realise that these ideas are all said in text in the body of the document, but a graphical representation of the idea has the potential to render this clearly, succinctly and compellingly and I think is worth exploring. (I realise I am treading a fine line here as a reviewer, potentially straying into editorial comments, but as the tone of the article is didactic as a reviewer I would like to see a clear visual representation of the information conveyed within the body text.)


The authors seem to imply that there is a ‘true’ measure of laterality that the fMRI paradigms are trying to reveal. If laterality does vary as a function of the baseline task, which is the ‘true’ laterality, or which is the ‘clinically relevant’ laterality? Can we even know this? Implicit in some of the discussion in the paper is the idea that strong laterality is a marker of a good task, but is this really the case? If we are seeking the ‘truth’ of laterality for a given region or task, then selecting the task that has the strongest laterality might be biasing the results.
This really comes down to the question of what are we hoping to obtain through our laterality measurements? Is reliability/stability more important than strength of laterality? Or is it important, for instance, to identify what degree of laterality is indicative of redundancy of function, such that unilateral resection would be safe?
I think the Discussion would benefit from some consideration of “how do we know that we have accurately captured laterality of a region/task?”
At the least this would make overt the idea that the strongest lateralisation is the best. I admit though that this is a difficult question to answer. I just worry that the ‘strongest lateralisation is best’ approach runs the risk of relying on the metric do dictate meaning, rather than having the underlying functionality of an area dictate meaning.
Followed to the extreme, we may continue refining tasks for a given region until we come up with one that yields very strong lateralisation, ignoring those with weaker lateralisation that we tested in earlier iterations. If followed to this extreme, have we then potentially lost important information about the non-dominant hemisphere (that resecting it might cause a deterioration in function; or that resecting the dominant hemisphere might not be as disastrous as feared because the non-dominant hemisphere could support residual function)?
These are difficult questions to answer – I don’t pretend to have the answers to them. I am just urging caution with the ‘strongest LI is best’ approach, and consideration of the question regarding functional consequence of laterality measures (the difficulty of evaluating the relative utility of different LIs obtained via different tasks). It seems to me that one of the best ways to answer these questions is with surgical outcome studies. And once viewed from this light, the relevant metric would be the task for which the LI is best predictor of outcome, regardless of whether this is also the task with the strongest LIs.

===============================

In summary, I found this manuscript to be a useful summary of the literature, raising some important issues in the field. If some of the above comments could be addressed I believe this will increase the utility of the paper.

·

Basic reporting

The manuscript is very well written in Standard English and there are no issues with the basic reporting. Literature references and context seem to be in order. The standard article structure for systematic reviews is followed. The article does not seem to have a clear hypothesis, though, that is something the authors could potentially work on.

Experimental design

As far as I can judge it, the authors followed all standards for such studies. I do not see any issues with the design of this systematic review. The research question is well defined and meaningful. Technical standards are fulfilled and methods description is sufficient enough for potential replication.

Validity of the findings

This seems to be a very robust and statistically sound finding, and conclusions are in line with the data.

Additional comments

This is an interesting systematic reviews that covers an important question in the literature. The authors should work on clearly stating a hypothesis, but apart from that I see no problems with accepting this work.

---

## Round 0.2 · accepted · Accept

I am happy to inform you that your manuscript has been accepted for publication on PeerJ.

·

Basic reporting

ok

Experimental design

ok

Validity of the findings

ok

Additional comments

I don't see any remaining issue with the paper and think it could be recommended for acceptance now.